# Factors Related to Beliefs about Medication in Ischemic Stroke Patients

**DOI:** 10.3390/jcm11133825

**Published:** 2022-07-01

**Authors:** Gye-Gyoung Kim, Sung-Hee Yoo, Man-Seok Park, Hyun-Young Park, Jae-Kwan Cha

**Affiliations:** 1College of Nursing, Chonnam National University, Gwangju 61469, Korea; niwayakim@hanmail.net; 2Department of Neurology, Chonnam National University Hospital, Gwangju 61469, Korea; mspark@chonnam.ac.kr; 3Department of Neurology, Wonkwang University Hospital, Iksan 54538, Korea; hypppark@hanmail.net; 4Department of Neurology, Dong-A University Hospital, Busan 49201, Korea; nrcjk65@gmail.com

**Keywords:** belief, medication, stroke, medication adherence

## Abstract

Background: Medication beliefs are known as predictors of medication adherence. However, understanding of the relevance of these beliefs is lacking. Therefore, this study aimed to identify medication beliefs, and their influencing factors, in stroke survivors. Methods: This was a secondary analysis, using baseline data from a longitudinal study conducted to predict long-term medication adherence in Korean stroke survivors, and included 471 patients. Medication beliefs were investigated using the Belief about Medicine Questionnaire (BMQ), and the belief score and attitudes were derived from the “necessity” and “concern” scores, which are subscales of the BMQ. Multiple linear regression was used to determine independent factors influencing medication beliefs. Results: The mean score of medication beliefs is 7.07 ± 6.32, and the accepting group comprises 288 patients (61.1%). Medication beliefs are significantly higher in older adults (*p* < 0.001), females (*p* = 0.001), and patients with non-type D personality (*p* = 0.023), low-state anxiety (*p* < 0.001), high stroke severity (*p* = 0.001), a high number of medications (*p* < 0.001), and high knowledge about medications (*p* = 0.001). Conclusion: This study shows that type D personality, state anxiety, and knowledge about medication are major mediating factors for improving medication beliefs. These results may help healthcare professionals develop strategies to enhance medication adherence, by improving patients’ medication beliefs.

## 1. Introduction

Stroke is a leading cause of serious disability and death in Korea, and is a disease with a high risk of recurrence. The annual direct social costs of stroke are approximately KRW 1.684 trillion [1], necessitating efforts to reduce its recurrence. The use of antithrombotic drugs, revascularization, and risk factor control are recommended to prevent the recurrence of ischemic stroke, including transient ischemic attacks (TIA) [2]. Therefore, medication adherence, which means regular use of antithrombotic drugs and risk factor management drugs, is essential for ischemic stroke prevention.

However, medication adherence remains low at 64.1% in stroke survivors [3], and 46.7% in patients with established atherosclerotic disease [4]. Non-adherence is shown to result in increased vascular events such as strokes, myocardial infarction, and all-cause mortality [4]. Therefore, multifaceted efforts are required to improve medication adherence of stroke patients who require continuous medication and management.

Beliefs about medication were studied as a predictive factor for medication adherence, and a representative factor that can be influenced by medical personnel [5,6,7]. Horne reports that the necessity of, and concerns about, medication determines participants’ belief in medical treatment, which in turn affects medication adherence [8]. When making judgments about medication, patients have concerns about adverse effects, dependence, and harm, and decide whether medication is a necessity in spite of these possibilities [9]. For this reason, it is important to identify factors that influence patients’ medication concerns, and those that influence their perception of the medication’s necessity. However, while previous studies examine beliefs about medication as an antecedent factor affecting medication adherence [6,7], few studies investigate factors that affect beliefs about medication [10,11,12], especially in stroke patients.

In addition, an integrative review study conducted among cancer patients on the influencing factors of medication beliefs shows that the number of prescription drugs, the frequency or quality of communication with the healthcare provider, the degree of information, and low physical functioning are related to the perception of “necessity” for medication, while being female, having depression, and having negative experiences with drugs are related to “concerns” [10]. However, in previous studies, these socio-economic factors, disease and therapy-related factors, and psychological factors are only partially considered in each study as influencing beliefs about medication. In addition, it is necessary to investigate how the necessity and concern about medications differ according to stroke severity in stroke patients, due to the substantial difference in physical function resulting from stroke sequelae.

Therefore, the purpose of this study was to ascertain beliefs about medication in ischemic stroke patients, and determine whether there is a difference in these belief attitudes depending on stroke severity. Considering various potential factors, we also attempted to identify not only factors that affect medication beliefs, but also factors influencing both beliefs about necessity and the concern of patients, which are sub-concepts of medication beliefs.

## 2. Methods

### 2.1. Study Design and Subjects

This study was a secondary analysis to identify medication beliefs and their influencing factors, using only baseline data from a primary study, which was a prospective, multicenter, longitudinal study conducted to identify long-term medication adherence and their predictors in Korean ischemic stroke survivors. As the primary study was to determine the natural course of drug use for 1 year after discharge in ischemic stroke patients, all patients diagnosed with their first-ever ischemic stroke, including TIA, were included, regardless of cognitive or language impairment. Out of a total of 600 patients included in the primary study, baseline data from 471 patients who responded to a self-report questionnaire on various psychological variables, including their beliefs about medication, were included in this secondary analysis.

### 2.2. Measurement

We examined some of the variables of socio-economic, psychological, condition-related, therapy-related, and healthcare team-related characteristics from the primary study. The socio-economic characteristics included age, sex, education level, marital status, job type, and degree of social support. Patient-related psychological characteristics included type D personality, health literacy, state anxiety, and knowledge about stroke medication. Condition-related characteristics included stroke risk factors, such as hypertension or diabetic mellitus, stroke subtype (TIA or ischemic stroke), the mechanism of ischemic stroke by trial of org 10,172 in acute stroke treatment (TOAST) classification [13], and stroke severity according to the modified Rankin scale (mRS) [14] at discharge. In this study, stroke severity was classified into mRS 0–1 (the good outcome group), and mRS 2–5 (the poor outcome group) [15]. Therapy-related characteristics included the frequency, number, and type of stroke medication. Healthcare team-related characteristics included satisfaction with healthcare providers’ explanation about stroke and/or drugs. It was measured using one question on a five-point scale, “How satisfied are you with your healthcare providers’ explanation of diseases and drugs during hospitalization?”.

Medication belief as an outcome variable, social support, type D personality, health literacy, state anxiety, and knowledge about stroke medication were measured using the following tools.

### 2.3. Beliefs about Medicine Questionnaire: BMQ-Specific

Beliefs about medication were measured using the Beliefs about Medicines Questionnaire (BMQ) [9]. It consists of a total of ten questions, with five questions each on the “necessity” and “concerns” subscales. Each question is rated on a 5 point Likert scale. Total scores range from 5 to 25 for the two domains. The total belief score about medication is calculated as the difference between the necessity and concern scores, ranging from −20 to 20 points. In this study, each influencing factor was identified by deriving BMQ-necessity and BMQ-concern scores, as well as the total medication belief scores.

In addition, belief attitudes about medication were examined in this study. These attitudes were classified into 4 groups based on the 15 points for each of the necessity and concern subdomains: accepting (high necessity, low concern), ambivalent (high necessity, high concern), skeptical (low necessity, high concern), and indifferent (low necessity, low concern). We tried to determine the belief attitudes about medications in stroke survivors, and whether there was a difference in the attitude according to stroke severity (good vs. poor outcome). In this study, the Cronbach’s α of the necessity domain is 0.91, and that of the concern domain is 0.78.

### 2.4. ENRICHD Social Support Inventory (ESSI)

Social support was measured using the Korean version of the ENRICHD Social Support Inventory (ESSI) [16], which was used in the Enhancing Recovery in Coronary Heart Disease Patients (ENRICHD) study [17]. The ESSI consists of seven questions measured with a 5 point Likert scale, except for the last question, which is binary (yes/no). A higher total score indicates better social support. The Cronbach’s α of ESSI in this study is 0.86.

### 2.5. Type D Personality Scale-14: DS14

Type D personality was measured using the Type D Personality Scale-14 (DS14) [18]. Type D personality is referred to as the “distressed” personality type, where individuals tend to experience negative emotions, usually due to a sad and depressing outlook on life, and hide their emotions for fear of how others will react. The DS14 consists of 14 questions with 2 subscales, “negative affectivity” and “social inhibition.” Each question is rated on a 5 point Likert scale ranging from “strongly disagree” (0 points) to “strongly agree” (4 points). A score of 10 points or higher on both subscales indicates that the respondent has a type D personality. In this study, Cronbach’s α for both negative affectivity and social inhibition is 0.86.

### 2.6. Health Literacy

Health literacy was measured using the Rapid Estimate of Adult Literacy in Medicine-Short Form (REALM-SF), which consists of seven items testing the respondents’ vocabulary comprehension [19]. Patients who understood the word clearly enough to explain its meaning were given 1 point, whereas those who vaguely perceived the meaning, only heard the word, or did not know the meaning of the word were given 0 points. A higher total score indicates better health literacy. The Cronbach’s α in this study is 0.94.

### 2.7. State-Trait Anxiety Inventory (STAI)

State anxiety, which reflects the patient’s current level of anxiety, was measured using the Korean version of Spielberger’s State-Trait Anxiety Inventory (STAI) subscale [20]. It consists of 20 questions on a 4 point Likert scale, with a total score ranging from 20 to 80. A higher score indicates higher levels of anxiety. The Cronbach’s α in this study is 0.93.

### 2.8. Knowledge about Stroke Medication

Knowledge about stroke medication was measured with a tool developed by the researcher. Based on Korean clinical practice guidelines for stroke [21], seven questions, evaluating respondents’ knowledge concerning the persistence of antithrombotic drugs, recognition of adverse effects, coping methods, etc., were developed. This tool was verified with content validity by five experts, and face validity by ten stroke patients. Each question could be answered with “yes”, “no”, or “not sure”, and only 1 point was awarded for each correct answer. Higher scores indicate higher knowledge about stroke medication.

### 2.9. Data Collection

All data were obtained from the primary study for subjects who met this study’s purposes. Baseline data in the primary study were collected through face-to-face interviews with trained research nurses at each center, between September 2017 and March 2018, from three regional stroke centers in South Korea. Some data on condition- and therapy-related factors were collected from electronic medical records (EMRs). The data collection and all procedures of the primary study adhered to the principles outlined in the Declaration of Helsinki [22], and Institutional Review Board approval for this secondary analysis was obtained along with the primary study.

### 2.10. Statistical Analysis

Data were presented in number and percentage, or mean and standard deviation. The participants’ medication belief scores were presented using scatter plots, and the distributions of belief attitudes in the four aforementioned groups were presented as numbers and percentages. The difference in belief attitudes about medication between the good (mRS 0–1) and poor (mRS 2–5) outcome groups was verified by Fisher’s exact test.

Independent *t*-tests, one-way ANOVA, and Pearson’s correlation were used to examine factors related to medication necessity, concerns, and total belief scores, respectively. Stepwise multiple linear regression was performed to identify independent factors influencing the overall beliefs and belief subscales.

SPSS version 23.0 (SPSS Inc., Chicago, IL, USA) for Windows was used for statistical analyses. A two tailed *p* < 0.05 was considered to indicate a significant difference.

## 3. Results

### 3.1. Participants’ Medication Beliefs and Belief Attitudes

Out of a total of 471 patients, the mean age is 63.42 ± 12.16 years, and 147 (31.2%) patients are women (Table 1). The mean necessity score is 19.36 ± 4.18, the mean concern score is 12.29 ± 4.17, and the mean medication belief score is 7.07 ± 6.32. Among a total of 471 patients, 288 patients (61.1%) are the most as an accepting group, and 14 patients (3.0%) are the least as a skeptical group. Compared to the mRS 0–1 group, the accepting group (73.4% vs. 55.2%) is larger in the mRS 2–5 group, while the ambivalent (21.4% vs. 30.9%), indifferent (3.9% vs. 10.1%), and skeptical (1.3% vs. 3.8%) groups are smaller. This difference is statistically significant (*p* = 0.001) (Figure 1).

### 3.2. Factors Related to Necessity, Concerns, and Medication Beliefs

Univariate analysis was performed to find factors associated with each of the necessity, concern, and overall medication beliefs (Table 1). To identify the independent factors that influence each of these, variables that are statistically significant in the univariate analysis are subsequently inserted into the multiple regression model, except for medication types that overlapped with vascular risk factors and TOAST, which applies only to ischemic stroke patients.

As a result, age (*p* < 0.001), gender (*p* = 0.001), type D personality (*p* = 0.023), state anxiety (*p* < 0.001), stroke severity (*p* = 0.001), number of medications (*p* < 0.001), and knowledge about medications (*p* = 0.001) are significantly associated with overall medication beliefs (Table 2). Specifically, patients with higher age, stroke severity, number of medications at discharge, and knowledge about stroke medications have higher beliefs about medications. Conversely, male patients with type D personality, or high levels of state anxiety, have lower beliefs about medications.

Concerns about medications are significantly higher in young (*p* = 0.005), educated (*p* = 0.008) patients with type D personality (*p* < 0.001), those with high-state anxiety (*p* = 0.001), no diabetes (*p* = 0.001), hyperlipidemia (*p* = 0.004), low stroke severity (*p* = 0.026), and low medication knowledge (*p* = 0.008) (Table 2). The Durbin–Watson value is 1.819, which is close to 2, indicating that there is no autocorrelation. The tolerance is 0.67–0.97, and the variance inflation factor (VIF) is less than 10, indicating that there are no problems with collinearity.

## 4. Discussion

This study was performed to ascertain beliefs and belief attitudes about medication in ischemic stroke patients, including TIA, and to identify factors affecting necessity, concern, as well as overall medication beliefs. As a result, the mean medication belief score is 7.07, and approximately 61% of patients belong to the accepting group. Patients of an older age, who are female, with higher stroke severity, a higher number of medications, higher knowledge about medication, and without type D personality and anxiety have a stronger belief in medication. In particular, despite being young, educated, and having low stroke severity, patients with type D personality and anxiety, as well as lack of knowledge about medication, have a significantly higher level of concern about medication.

The mean medication belief score in our study is 7.1 (a necessity score of 19.4 and a concern score of 12.3). This is slightly lower than the mean belief score of 7.9 (a necessity score of 21.0 and a concern score of 13.1) for kidney transplant patients using immunosuppressant medication [23], but is significantly higher than 4.0 (a necessity score of 17.6 and a concern score of 13.6), and 5.3 (a necessity score of 18.6 and a concern score of 13.3) for hypertensive [24] and Parkinson’s patients [12], respectively. We believe these differences are based on the severity of each disease, with a higher severity raising awareness of the necessity for medication, which in turn leads to stronger belief in the efficacy of medications. Even among ischemic stroke patients, this study shows that the more severe the disability, the higher the belief in the medication. In fact, the acceptance rate of medication is 73.4% in the mRS 2–5 group, which is considerably higher than that of the mRS 0–1 group (55.2%). However, the rate of indifference with low awareness of necessity and concern about medication is 10.1% in the mRS 0–1 group, which is higher than that of the mRS 2–5 group (3.9%). These results are consistent with previous studies that find that the higher the cancer stage in cancer patients [10], and the higher Hoehn and Yahr stage in Parkinson’s patients [12], the higher the perceived necessity for and beliefs about medication.

Age is a strong factor affecting medication belief, especially for older adults. This may be because older patients are more likely to have various chronic diseases than younger patients, and tend to be more dependent on treatments or medication [25]. Our study supports this supposition, by showing higher medication necessity and belief in patients with some stroke risk factors and higher medication numbers. Consistent with a previous study, women have a stronger belief in medication than men [10]. However, older female patients have a high risk of multidrug prescriptions and lack of knowledge about medications, which may lead to misuse of drugs [26,27]. Thus, even though they have a strong belief in the effectiveness of medication, we need to help them obtain the right knowledge about the drugs they are taking.

The results show that, similar to prior research, high knowledge about medications is associated with high medication beliefs [10,28]. In particular, low knowledge shows a significant correlation with “concern” about medication, and high satisfaction with health providers’ explanation shows strong awareness of the “necessity” of medication. These results confirm that establishing an educational relationship with health providers is important in enhancing medication belief.

Interestingly, in our study, higher educational level is not associated with increased medication belief. One possible explanation is that the older patients included in our study are less educated and, despite their low knowledge of medication, their concerns about medication are significantly lower, thus, leading to stronger belief. In other words, younger patients with higher education levels are unable to trust the medications completely, either because they are unimpressed by their effects, or have significant concerns. This finding newly demonstrates younger stroke patients’ need for education to reduce anxiety and concerns. and emphasize the necessity of medication.

We also measured type D personality and state anxiety to consider patients’ psychological characteristics. The results show that patients with higher levels of anxiety have a lower perception of the necessity for medication, increased concern, and a lower overall belief in medication. A similar result is found in patients with type D personality, who are known to be vulnerable to negative emotions such as depression, anxiety, and stress, and tend to withhold self-expression in social interactions [18]. In this study, patients with type D personality are significantly more anxious about their condition (*p* = 0.001). Therefore, these two factors might be associated with increased medication concern, and decreased belief. Indeed, because type D personality is known to have a negative effect on medication adherence [29], we need to further explore strategies to improve medication adherence, by improving patients’ beliefs, considering stroke patients’ personality type [30].

This study is a secondary analysis of some baseline data from a primary study conducted to investigate long-term medication persistence and adherence, and their predictive factors, in acute ischemic stroke patients. Therefore, there is a limitation in that variables known to be related to medication beliefs in previous studies may have been omitted. However, since we consider various psychological and therapy-related characteristics, as well as socio-economic and disease-related characteristics of stroke patients, we believe that almost all of the various potential variables related to medication beliefs are addressed in this study. In addition, this study was conducted in some stroke centers in Korea, so there may be limitations in terms of generalization.

## 5. Conclusions

This study confirms that patients who are younger, less disabled, and less knowledgeable about medication have a lower belief in medication. In addition, it is identified that patients’ psychological characteristics, such as type D personality and anxiety state, can influence their medication beliefs. Therefore, health care providers need to evaluate not only patients’ age, sex, and stroke severity, but also psychological factors such as personality and anxiety. Therefore, it is believed that various strategies that provide information enabling patients to fully recognize the necessity of medication, despite concerns about drug side effects, will increase their belief about medication, and further improve their medication adherence. These interventions may need to be initiated during hospitalization, and continued following discharge, in order to improve long-term medication adherence in stroke survivors.

In the future, the development of interventions based on theoretical models to improve patients’ knowledge, belief about medication, and their internal and external motivation are required. Further studies need to be conducted to determine whether this changes the health-promoting behavior of medication adherence.

## Figures and Tables

**Figure 1 jcm-11-03825-f001:**
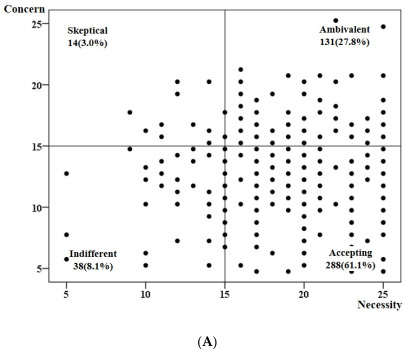
Belief attitudes about medication in ischemic stroke patients. (**A**) Shows belief attitudes towards medication of all stroke patients (*n* = 471). (**B**) Shows the differences in the belief attitudes about medication between mRS 0–1 group (*n* = 317) and mRS 2–5 group (*n* = 154), which are significantly different (*p* = 0.001).

**Table 1 jcm-11-03825-t001:** Factors related to necessity, concerns, and medication beliefs by univariate analysis.

(*n* = 471)											
Characteristics		Medication Beliefs									
		*n* (%) or M ± SD	Necessity			Concerns			Total score		
			M ± SD	*t* or F or r	*p*	M ± SD	*t* or F or r	*p*	M ± SD	*t* or F or r	*p*
Socio-economic factors											
Age (year)		63.42 ± 12.16		0.30	<0.001		−0.26	<0.001		0.37	<0.001
Gender	Male	324 (68.8)	18.86 ± 4.25	−3.89	<0.001	12.65 ± 3.95	2.72	0.007	6.21 ± 5.87	−4.24	<0.001
	Female	147 (31.2)	20.46 ± 3.82			11.48 ± 4.52			8.98 ± 6.86		
Education level											
Primary school and under ^a^		138 (29.3)	20.67 ± 3.76	7.55	<0.001	11.04 ± 4.72	6.19	<0.001	9.63 ± 6.93	12.17	<0.001
Middle school ^b^		69 (14.6)	19.45 ± 3.51		a > c,d ^††^	12.67 ± 3.75		a < b,c,d ^††^	6.78 ± 5.83		a > b,c,d ^††^
High school ^c^		165 (35.0)	18.79 ± 3.98			12.69 ± 3.86			6.10 ± 5.25		
Above college ^d^		99 (21.0)	18.42 ± 5.01			13.08 ± 3.80			5.34 ± 6.40		
Marital status	Married ^a^	375 (79.6)	19.16 ± 4.12	6.75	0.001	12.50 ± 4.00	11.65	<0.001	6.66 ± 5.98	16.09	<0.001
	Widowed ^b^	61 (13.0)	21.11 ± 3.77		b > a,c ^†^	10.11 ± 4.78		a,c > b ^†^	11.00 ± 7.08		b > a,c ^†^
	Single ^c^	35 (7.4)	18.49 ± 4.77			13.83 ± 3.68			4.66 ± 5.94		
Job	Employed	270 (57.3)	18.69 ± 4.28	−4.14	<0.001	12.94 ± 3.86	3.93	<0.001	5.74 ± 5.69	−5.32	<0.001
	Unemployed	201 (42.7)	20.27 ± 3.87			11.41 ± 4.41			8.86 ± 6.69		
ESSI: social support		20.96 ± 3.90		−0.09	0.045		−0.13	0.006		0.02	0.641
Psychological factors											
Health literacy		3.62 ± 2.87		−0.001	0.982		0.16	0.001		−0.11	0.022
D type	Yes	235 (49.9)	19.79 ± 3.69	2.24	0.025	13.59 ± 4.01	7.13	<0.001	6.20 ± 5.81	−3.02	0.003
personality	No	236 (50.1)	18.93 ± 4.58			10.99 ± 3.92			7.94 ± 6.69		
STAI: state anxiety		43.74 ± 11.71		−0.14	0.002		0.16	0.001		−0.20	<0.001
Knowledge about medication		4.98 ± 1.87		0.05	0.289		−0.16	<0.001		0.14	0.002
Condition-related factors											
Stroke risk factors ^§^											
HTN	Yes	301 (63.9)	19.72 ± 4.21	2.53	0.012	11.77 ± 4.31	−3.79	<0.001	7.96 ± 6.55	4.29	<0.001
	No	170 (36.1)	18.72 ± 4.05			13.21 ± 3.74			5.51 ± 5.58		
DM	Yes	141 (29.9)	19.88 ± 4.14	1.76	0.078	11.32 ± 4.13	−3.33	0.001	8.56 ± 6.59	3.37	0.001
	No	330 (70.1)	19.14 ± 4.18			12.70 ± 4.12			6.44 ± 6.11		
Hyperlipidemia	Yes	191 (40.6)	19.94 ± 3.80	2.49	0.011	13.09 ± 4.23	3.49	0.001	6.85 ± 6.19	−0.64	0.522
	No	280 (59.4)	18.97 ± 4.38			11.74 ± 4.05			7.23 ± 6.42		
MI	Yes	47 (10.0)	20.57 ± 3.72	2.11	0.036	11.60 ± 4.35	−1.20	0.232	8.98 ± 6.07	2.19	0.029
	No	424 (90.0)	19.23 ± 4.21			12.36 ± 4.15			6.86 ± 6.32		
Afib	Yes	53 (11.3)	20.38 ± 4.41	1.89	0.060	13.36 ± 4.90	1.72	0.090	7.02 ± 7.23	−0.07	0.946
	No	418 (88.7)	19.23 ± 4.14			12.15 ± 4.05			7.08 ± 6.21		
Current	Yes	151 (32.1)	18.07 ± 4.26	−4.70	<0.001	12.50 ± 3.44	0.85	0.396	5.57 ± 5.50	−3.83	<0.001
smoking	No	320 (67.9)	19.97 ± 4.01			12.18 ± 4.48			7.78 ± 6.57		
Stroke subtype (*n* = 442)	Ischemic	433 (91.9)	19.47 ± 4.23	1.90	0.058	12.19 ± 4.20	−1.71	0.088	7.28 ± 6.40	3.00	0.004
	TIA	38 (8.1)	18.13 ± 3.42			13.39 ± 3.68			4.74 ± 4.87		
TOAST (*n* = 433)	LAA ^a^	104 (22.1)	19.39 ± 4.48	5.09	0.002	11.52 ± 4.19	3.55	0.015	7.88 ± 6.95	6.18	<0.001
	CE ^b^	63 (13.4)	19.97 ± 4.22		d > c ^†^	12.83 ± 4.79			7.14 ± 6.47		d > c ^††^
	SVO ^c^	193 (41.0)	18.79 ± 3.93			12.69 ± 3.90			6.09 ± 5.69		
	Others ^d^	73 (15.5)	20.95 ± 4.26			11.26 ± 4.21			9.68 ± 6.64		
mRS at discharge		1.31 ± 1.27		0.19	<0.001		−0.19	<0.001		0.24	<0.001
mRS subgroup	mRS 0–1	317 (67.3)	18.85 ± 4.26	3.87	<0.001	12.73 ± 4.02	−3.36	0.001	6.12 ± 5.94	4.83	<0.001
	mRS 2–5	154 (32.7)	20.42 ± 3.82			11.37 ± 4.33			9.05 ± 6.64		
Therapy-related factors											
Frequency of medication		2.28 ± 0.58		0.18	<0.001		0.04	0.453		0.09	0.041
Number of medications		6.38 ± 2.55		0.29	<0.001		−0.02	0.632		0.20	<0.001
Type of stroke medicine ^§^											
Anti-platelet	Yes	416 (88.3)	19.21 ± 4.19	−2.21	0.027	12.16 ± 4.09	−1.77	0.078	7.04 ± 6.28	−0.29	0.770
	No	55 (11.7)	20.53 ± 3.96			13.22 ± 4.69			7.31 ± 6.70		
Warfarin	Yes	13 (2.8)	20.15 ± 4.02	0.69	0.488	12.85 ± 4.81	0.49	0.624	7.31 ± 6.92	0.14	0.893
	No	458 (97.2)	19.34 ± 4.19			12.27 ± 4.16			7.07 ± 6.31		
NOAC	Yes	56 (11.9)	20.41 ± 4.34	2.01	0.045	13.13 ± 4.70	1.61	0.109	7.29 ± 6.43	0.27	0.790
	No	415 (88.1)	19.22 ± 4.14			12.17 ± 4.09			7.05 ± 6.32		
Anti-HTN	Yes	250 (53.1)	19.69 ± 4.11	1.81	0.071	11.68 ± 4.25	−3.40	0.001	8.01 ± 6.34	3.45	0.001
agent	No	221 (46.9)	18.99 ± 4.23			12.97 ± 3.98			6.02 ± 6.14		
Anti-diabetic	Yes	123 (26.1)	19.82 ± 4.21	1.42	0.156	11.24 ± 3.86	−3.29	0.001	8.59 ± 6.41	3.11	0.002
agent	No	348 (73.9)	19.20 ± 4.16			12.66 ± 4.22			6.54 ± 6.21		
Anti-lipidemic	Yes	451 (95.8)	19.37 ± 4.17	0.12	0.904	12.25 ± 4.13	−0.78	0.435	7.11 ± 6.34	0.60	0.552
agent	No	20 (4.2)	19.25 ± 4.60			13.00 ± 4.96			6.25 ± 6.01		
Healthcare team-related factors											
Satisfaction with healthcare providers’ explanation		3.98 ± 0.91		0.12	0.008		−0.05	0.248		0.12	0.012

Abbreviations: BMQ, Beliefs about Medicines Questionnaire; ESSI, ENRICHD Social Support Inventory; STAI, State-Trait Anxiety Inventory; HTN, hypertension; DM, diabetes mellitus; MI, myocardial infarction; Afib, atrial fibrillation; TIA, transient ischemic attack; TOAST, trial of ORG 10,172 in acute stroke treatment; LAA, large-artery atherosclerosis; CE, cardioembolism; SVO, small-vessel occlusion; mRS, modified Rankin scale; NOAC, new oral anticoagulant. Notes. ^†^ Scheffe test; ^††^ Games–Howell test; ^§^ multiple response.

**Table 2 jcm-11-03825-t002:** Factors influencing medication beliefs of ischemic stroke patients.

(*n* = 471)												
Characteristics	Medication Beliefs											
	Necessity				Concerns				Total score			
	B	S.E.	β	*p*	B	S.E.	β	*p*	B	S.E.	β	*p*
Age	0.08	0.02	0.23	<0.001	−0.05	0.02	−0.14	0.005	0.15	0.02	0.28	<0.001
Male	−1.28	0.38	−0.14	0.001					−1.87	0.56	−0.14	0.001
Education level: Primary school and under					−1.17	0.44	−0.13	0.008				
Type D personality	0.79	0.36	0.10	0.028	1.94	0.36	0.23	<0.001	−1.22	0.53	−0.10	0.023
State anxiety	−0.04	0.02	−0.12	0.005	0.05	0.02	0.14	0.001	−0.09	0.02	−0.1	<0.001
Diabetes mellitus					−1.29	0.38	−0.14	0.001				
Hyperlipidemia					1.04	0.36	0.12	0.004				
Stroke severity according to mRS	0.34	0.15	0.10	0.021	−0.32	0.14	−0.10	0.026	0.72	0.21	0.14	0.001
Number of medications	0.39	0.07	0.24	<0.001					0.44	0.10	0.18	<0.001
Knowledge about medication					−0.26	0.10	−0.11	0.008	0.47	0.14	0.14	0.001
	*R*^2^ (Δ*R*^2^) = 0.46 (0.209), adjusted *R*^2^ = 0.20, F = 20.20, *p* < 0.001				*R*^2^ (Δ*R*^2^) = 0.48 (0.234), adjusted *R*^2^ = 0.22, F = 17.46, *p* < 0.001				*R*^2^ (Δ*R*^2^) = 0.52 (0.268), adjusted *R*^2^ = 0.26, F = 24.07, *p* < 0.001			

Abbreviations: S.E, standard errors; mRS, modified Rankin scale. Notes. The symbol B indicates the unstandardized coefficient, and the symbol β indicates the standardized coefficient.

## Data Availability

Due to the sensitive nature of the questions asked in this study, survey respondents were assured raw data would remain confidential and would not be shared. Data not available/the data used is confidential.

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
