# Peer review of "Factors Related to Beliefs about Medication in Ischemic Stroke Patients"

_jcm, 2022, doi:10.3390/jcm11133825_

Round 1
Reviewer 1 Report
This is a very interesting research article. Because the patient's various factors affect the confidence of medication, it determines the therapeutic effect of many diseases. Therefore, this study has certain significance and has good guiding significance for clinicians to reasonably evaluate medication compliance.
1、In Figure 1, there are some quantified tables of the number of cases on the coordinate table of 4 quadrants. How to analyze and process these data?
2、.The symbol β in Table 2, But in note refer a B? Please confirm.
3、 Explain the concept of secondary analysis.
4、What is D personality, please give detailed information.
5、Adherence remains low of medication is a phenomenon which seriously affects the effect of medication adherence. Through the analysis of a large number of cases, it was found that the most important reason for the low dependence on drugs in clinical practice, what are the important tips for doctors to use drugs and after discharge?
6、When describing data results, a more formal language should be used instead of a simple one. For example, As a result, older age (p<.001), female (p=.001), non-type D personality (p=.023), lower state nxiety (p<.001), higher stroke severity (p=.001), higher number of medications (p<.001)....
7、The language expression in this manuscript needs to be further polish and its grammatical errors revised.
Author Response
Thank you for the thoughtful comments and suggestions. We have corrected the manuscript accordingly and presented our answers to the comments in plain italics. The revised text is shown in red. The entire revised manuscript was edited for grammar and language.

Reviewer 2 Report
The study presents an analysis of stroke patients' beliefs about medication based on a previous large study carried out in Korea. The analyses are well carried-out and the presentation is clear. Finally, the limitations of the study are also clearly defined.
The authors examine in depth the beliefs of patients about the necessity and concerns about medication, a key issue in stroke management. In further research, it would be interesting to consider this question from the perspective put forward by Bandura and colleagues about the role of perceived self-effectiveness in health management. In particular, his social cognitive theory proposes an articulated structure in which self-efficacy beliefs operate together with goals, outcome expectations, and perceived environmental impediments and facilitators in the regulation of human motivation, behavior, and well-being, thus influencing health functioning. Placing the important question of medication within an articulated theoretical framework might significantly enhance our understanding of the phenomenon and foster the possibility of active intervention. These considerations do not detract from the interest in the current research but simply aim to foster further research along these lines.
Author Response
We appreciate the reviewer’s positive evaluation of this study and helpful suggestions for future directions. We have proposed interventions to promote medication adherence in the Information-Motivation-Behavioral skill (IMB) model, and intervention research is in progress. This is based on the primary study on the predictors of long-term medication persistence and adherence, as well as the secondary analysis on the understanding of medication beliefs as major predictors of medication adherence. We believe that your proposed Bandura’s social cognitive theory contains concepts similar to the IMB model. We completely agree that it is important to develop interventions based on this theoretical framework. Thus, the reviewer’s suggestions for further research were added to the “Conclusion” section as follows (page 10).
- “In the future, the development of interventions based on theoretical models that improve patients’ knowledge, belief about medication, and their internal and external motivation are needed. Further studies should be undertaken to determine whether this changes the health-promoting behavior of medication adherence.”